# The metabolic profile of waist to hip ratio–A multi-cohort study

**Lars Lind[1]\*, Shafqat Ahmad[1], Sölve Elmståhl[2], Tove Fall[1]**

**1** Department of Medical Sciences, Uppsala University, Uppsala, Sweden, **2** Division of Geriatric Medicine, Department of Clinical Sciences in Malmö, Lund University, Malmö, Sweden

\* lars.lind@medsci.uu.se

## Abstract

### Background

The genetic background of general obesity and fat distribution is different, pointing to separate underlying physiology. Here, we searched for metabolites and lipoprotein particles associated with fat distribution, measured as waist/hip ratio adjusted for fat mass (WHRadjfatmass), and general adiposity measured as percentage fat mass.

### Method

The sex-stratified association of 791 metabolites detected by liquid chromatography–mass spectrometry (LC-MS) and 91 lipoprotein particles measured by nuclear magnetic spectroscopy (NMR) with WHRadjfatmass and fat mass were assessed using three population-based cohorts: EpiHealth (n = 2350) as discovery cohort, with PIVUS (n = 603) and POEM (n = 502) as replication cohorts.

### Results

Of the 193 LC-MS-metabolites being associated with WHRadjfatmass in EpiHealth (false discovery rate (FDR) <5%), 52 were replicated in a meta-analysis of PIVUS and POEM. Nine metabolites, including ceramides, sphingomyelins or glycerophosphatidylcholines, were inversely associated with WHRadjfatmass in both sexes. Two of the sphingomyelins (d18:2/24:1, d18:1/24:2 and d18:2/24:2) were not associated with fat mass (p>0.50). Out of 91, 82 lipoprotein particles were associated with WHRadjfatmass in EpiHealth and 42 were replicated. Fourteen of those were associated in both sexes and belonged to very-large or large HDL particles, all being inversely associated with both WHRadjfatmass and fat mass.

### Conclusion

Two sphingomyelins were inversely linked to body fat distribution in both men and women without being associated with fat mass, while very-large and large HDL particles were inversely associated with both fat distribution and fat mass. If these metabolites represent a link between an impaired fat distribution and cardiometabolic diseases remains to be established.

**Data Availability Statement:** Due to Swedish laws on personal integrity and health data, as well as the decision by the Ethics Committee, we are not allowed to make any data including health variables open to the public, even if made anonymous. The data could be shared with other researchers after a

request to the steering committee (karl.
michaelsson@surgsci.uu.se).

**Funding:** The EpiHealth study was funded by the
Swedish Foundation for Strategic Research. POEM
and PIVUS were funded by Uppsala University
Hospital (ALF-grants). The funders had no role in
study design, data collection and analysis, decision
to publish, or preparation of the manuscript.

**Competing interests:** The authors have declared
that no competing interests exist.

## Introduction

Excess body adipose tissue could be analyzed in at least two dimensions, total fat mass and fat distribution. In epidemiology, increased fat mass is most often evaluated by use of body mass index (BMI), while fat distribution is assessed by either waist circumference (WC) or the waist/hip circumference ratio (WHR). Mendelian randomization studies have shown a positive causal effect of BMI [1, 2], as well as BMI-adjusted WHR [3], on cardiovascular risk factors and diseases, supporting the view that neither general obesity, nor a disadvantageous fat distribution are innocent phenomena.

Large-scale genetic studies are a way to search for pathophysiological pathways involved in obesity. Using data from several hundred thousand individuals have identified >250 genetic loci linked to BMI [4], and around 350 loci linked to WHR when adjusted for BMI [5]. In a review of the association between genetic loci and total fat mass and fat distribution, Fall et al. [6] pointed out that the biological pathways identified by these genes and further expression analyses are very different. For BMI, central nervous system pathways, with especially hippocampus, hypothalamus and the limbic system, plays a major role in terms of neurotransmission and energy balance. In contrast, genes related to WHR mainly reflects adipose tissue biology, insulin resistance and angiogenesis. Thus, genes regulating total fat mass and fat distribution points towards different mechanisms involved in these two dimensions of obesity. Another important finding is that for around one-third of the genetic loci linked to WHR a sex-interaction was seen, with generally stronger genetic effects in women compared to men.

Metabolomics is the study of small molecules (<1.5kD). A great number of studies have investigated the metabolomic profile of obesity, and a meta-analysis of 11 studies found high levels of branched-chain and aromatic amino acids, certain fatty acids and reduced levels of acylcarnitines and lysophosphatidylcholines to be the most common metabolic alterations in obese individuals [7]. There are also studies on the metabolomic profile of an altered fat distribution [8–13]. However, only a few studies have tried to disentangle if the metabolic profile of a disadvantageous fat distribution was different from that found in general obesity [14, 15].

A detailed description of the lipoprotein metabolic profile could be obtained by magnetic resonance spectroscopy (NMR) and a certain profile, with high levels of cholesterol in all VLDL and LDL subclasses and low levels in the larger classes of HDL together with elevated triglyceride levels in all lipoprotein classes except the largest classes of HDL, have been associated with cardiometabolic disease, such as myocardial infarction and stroke [16]. It is unclear if this lipoprotein profile is seen in subjects with a high WHR independently of general obesity.

The major aim of the present study is to investigate if the metabolic profile, assessed through LC-MS and lipoprotein profile measured through NMR, of a disadvantageous fat distribution is different from that found in general obesity, using a similar approach as in the genetic studies. Since we have measured fat mass by bioimpedance in the samples used in the present study, we adjusted WHR for fat mass percentage instead of BMI. Fat mass percentage is related to cardiovascular mortality [17], and all-cause mortality [18] independently of BMI, and is a more precise measure of the amount of adipose tissue than BMI. As the genetic studies of WHR adjusted for BMI points to sex-differences, we stratified the analyses of WHR by sex. In this study, we used one population-based study for discovery (EpiHealth) and a meta-analysis of another two studies as replication (PIVUS and POEM) in order to validate the findings in independent samples. The hypothesis tested was that we would find metabolites and lipoproteins being related to WHR independently of fat mass.

## Material and methods

### Population samples

**EpiHealth.** Starting 2011, a random sample of men and women in the age range 45 to 75 years were invited to a health screening survey, called EpiHealth, in the two Swedish cities Uppsala and Malmö [19]. In 2018, 25,000 individuals were included. Metabolomic data have been collected in a random subsample of 2,342 subjects attending the Uppsala site.

**POEM (Prospective investigation of Obesity, Energy and Metabolism).** The population-based POEM study is based on invitations to a random sample of 50-year old men and women living in Uppsala, Sweden [20]. Between Oct 2010 and Oct 2016, 502 individuals were included and metabolomics measurements have been performed in the total sample.

**PIVUS (Prospective Investigation of the Vasculature in Uppsala Seniors).** Between 2001 and 2004, 1,016 randomly selected men and women, all aged 70 years, were investigated [21]. They were all offered a new examination at the ages of 75 and 80. The present study use data from the 80-year examination in which metabolomics measurements have been performed in the total sample (n = 603).

All studies were approved by the Uppsala ethics committee (Application numbers 2010/402, 2011/045 and 2009/057) and all participants provided their informed written consent.

### Physical measurements and blood sampling

Waist circumference was measured at the umbilical level, while hip circumference was measured at the level of trochanter major. WHR is the ratio of those two measurements (waist/hip). Fat mass and body weight were assessed through a weight scale that also calculates fat mass by mean of bioimpedance (Tanita BC-418MA, Tokyo, Japan). Fat mass percentage is the fat mass divided by body weight and is the measurement used in the present study. Using both total body potassium and total body water [22], as well as dual-energy x-ray absorptiometry (DXA) [23] as comparative methods, measurement of fat mass with bioimpedance has been proven to be valid in previous studies. In addition, to validate that fat mass measured by bioimpedance is an accurate measurement of body fat in our setting, we compared the bioimpedance measurement with measurements with dual X-ray absorbmetry (DEXA, Lunar Prodigy, GE Healthcare) performed in 486 individuals in the POEM study. The Pearson´s correlation coefficient between these two measurements were 0.93.

The physical measurements were performed in the same fashion in all of the three studies.

Blood was drawn after an overnight fast in the POEM and the PIVUS cohorts, while 6 hours of fasting was required in EpiHealth. The blood was collected in EDTA tubes that were centrifuged and plasma was frozen in -80˚C for later analysis.

### Questionnaire

Life-style factors were evaluated using a questionnaire in all samples. In EpiHealth, leisure-time physical activity was assessed on a 5-level scale with 1 as sedentary and 5 as athlete training. Smoking was defined as years of smoking in life. Alcohol intake was assessed as drinks per week.

In the POEM and the PIVUS cohorts, leisure-time physical activity was assessed on a 4-level scale with 1 as sedentary and 4 as athlete training. Smoking variable was used as current smoking. Alcohol intake was not assessed in the PIVUS cohort at age 80 years neither in the POEM cohort. In all three cohort studies, education was defined on a three-level scale; <10, 10–12 and >12 years in school.

## Metabolomics

In all three study samples, metabolomics (Metabolon inc., USA) was performed on plasma samples being stored at -80˚C. 100μl of human plasma was utilized for analysis. 500μl of methanol was added to each sample. Samples were prepared using the automated MicroLab STAR® system from Hamilton Company. Several internal standards were added prior to the first step in the extraction process for QC purposes. To remove protein, dissociate small molecules bound to protein or trapped in the precipitated protein matrix, and to recover chemically diverse metabolites, proteins were precipitated with methanol under vigorous shaking for 2 min (Glen Mills GenoGrinder 2000) followed by centrifugation. The resulting extract was divided into five fractions: two for analysis by two separate reverse phases (RP)/UPLC-MS/MS methods with positive ion mode electrospray ionization (ESI), one for analysis by RP/ UPLC-MS/MS with negative ion mode ESI, one for analysis by hydrophilic interaction (HILIC)/UPLC-MS/MS with negative ion mode ESI, and one sample was reserved for backup. Metabolon's untargeted metabolomics panel utilizes a Waters ACQUITY ultra-performance liquid chromatography (UPLC) and a Thermo Scientific Q-Exactive or Q-Exactive Plus high resolution/accurate mass spectrometer interfaced with a heated electrospray ionization (HESI-I). The columns utilized were Waters BEH C18 2.1x100mm 1.7μm and Waters BEH Amide 2.1 x150mm 1.7μm.

Only annotated, non-xenobiotic metabolites with a detection rate >75% in all samples were used in the analyses (n = 791). The values were normalized and given in arbitrary units. The relative concentration of identified peaks associated with each chemical in the Metabolon library (where present), are obtained by measuring the area of the peak relative to the surrounding baseline. All peak areas are integrated for each biochemical, based on the authentic standard for each biochemical, providing a consistent quantitation of relative abundance.

Compounds were identified by comparison to library entries of purified standards or recurrent unknown entities. Metabolon maintains a library based on >3,000 authenticated standards that contains the retention time/index (RI), mass to charge ratio (*m/z*), and chromatographic data (including MS/MS spectral data) on all molecules present in the library. Metabolon has built and maintains a proprietary chemical library based on authentic standards that contains the retention time (RT), retention index (RI), mass to charge ratio (m/z), and mass spectral data (including MS/MS spectral data) on all molecules present in the library per method. Biochemical identifications are therefore based on three criteria: retention index within a narrow retention window of the proposed identification, accurate mass match to the library, and the MS/MS forward and reverse scores between the experimental data and authentic standards. The MS/MS scores are based on a comparison of the ions present in the experimental spectrum to the ions present in the library spectrum. While there may be similarities between molecules based on one of these factors, the use of all three data points can be used to accurately identify biochemicals.

Several types of controls were analyzed in concert with the experimental samples: a pooled matrix sample generated by taking a small volume of each experimental sample (or alternatively, use of a pool of well-characterized human plasma) served as a technical replicate throughout the data set; extracted water samples served as process blanks; and a cocktail of QC standards that were carefully chosen not to interfere with the measurement of endogenous compounds were spiked into every analyzed sample, allowed instrument performance monitoring and aided chromatographic alignment.

Internal standards (IS) were used for alignment of data and for QC of instrument performance. The IS were selected to span the chromatogram and allow the creation of an RI ladder. Furthermore, they were selected to be representative of the type of endogenous compounds

detected and therefore can be used to monitor consistency of chromatographic behavior and MS response.

Process standards are added during sample extraction to ensure consistent performance of the entire process from sample preparation through sample analysis.

**Instrument performance standards.** d35-octadecanoic acid, fluorophenylglycine, d5-indole acetate, chlorophenylalanine, Br-phenylalanine, d5-tryptophan, d4-tyrosine, d3-serine, d3-aspartic acid, d7-ornithine, d4-lysine.

**Process assessment standards.** Fluorophenylglycine, chlorophenylalanine.

Instrument variability was 5% as determined by calculating the median relative standard deviation (RSD) for the internal standards that were added to each sample prior to injection into the mass spectrometers. Overall process variability was 7% as determined by calculating the median RSD for all endogenous metabolites (i.e., non-instrument standards) present in 100% of the Client Matrix samples, which are technical replicates of pooled client samples. In a published comparison between the 4 MS platforms used, the average laboratory coefficient of variation (CV) on the 4 platforms was between 9.3 and 11.5%, average inter-assay CV ranged from 9.9% to 12.6% and average intra-assay CV ranged from 5.7% to 6.9% [24].

## Lipoprotein measurements

Lipoproteins and their content were quantified using high-throughput NMR metabolomics (Nightingale Health Ltd, Helsinki, Finland) [25]. The 14 lipoprotein subclass sizes were defined as follows: extremely large VLDL with particle diameters from 75 nm upwards and a possible contribution of chylomicrons, five VLDL subclasses, IDL, three LDL subclasses and four HDL subclasses. The following components of the lipoprotein subclasses were quantified: phospholipids (PL), triglycerides (TG), cholesterol (C), free cholesterol (FC), and cholesteryl esters (CE). Very few of the measurements of the extremely large VLDL were above the level of detection, so this subclass was not used in the further analysis in the present study.

Two NMR spectra were recorded for each plasma sample using 500 MHz NMR spectrometers (Bruker AVANCE IIIHD). The first spectrum is a presaturated proton spectrum, which features resonances arising mainly from proteins and lipids within various lipoprotein particles. The other spectrum is a Carr-Purcell-Meiboom-Gill T2-relaxation-filtered spectrum where most of the broad macromolecule and lipoprotein lipid signals are suppressed, leading to enhanced detection of low-molecular-weight metabolites. The identification and quantification used a company proprietary software (version 2020). Two internal control samples provided by the company were included in each 96-well plate for tracking the consistency over time. When the coefficient of variation (CV) was calculated based on these internal controls and duplicate samples, the mean CV was below 4%, and only a few metabolites showed a CV>10%.

## Statistics

All metabolites and lipoproteins were rank based inverse-normal transformed to obtain a normal distribution and the same mean level for each metabolite. Fat mass (percentage) and WHR were normally distributed.

Separate analyses were performed for the LC-MS metabolomics and NMR-based lipoprotein measurements.

For WHRadjfatmass, linear regression analyses were performed using metabolites as dependent variables and WHRadjfatmass as the independent variable. Potential confounders were used including fat mass percentage, age, education, smoking, alcohol, exercise habits and statin use (other antilipidemic agents are rarely used in Sweden). The same model was used in

all three study samples (except that alcohol was not included in the PIVUS and the POEM cohorts). Sex stratified analysis were performed for WHRadjfatmass. An interaction term between WHRadjfatmass and sex was used in a set of separate models to test the significance of any sex-interactions regarding the relationships between WHRadjfatmass and metabolites.

To be able to identify metabolites associated with WHRadjfatmass only, we also assessed the association of fat mass percentage with metabolites using linear regression models where metabolites were used as dependent variables and fat mass percentage as the independent variable. Confounders in the model were age, sex, education, smoking, alcohol, exercise habits and statin. The same model was used in all three study cohorts (except that data on alcohol intake was not available in the PIVUS and POEM samples).

EpiHealth was used as the discovery sample and a false discovery rate (FDR, Benjamin-Hochenberg) <0.05 was used to qualify metabolites to be evaluated in the validation step. The validation step was performed using results from a meta-analysis (inverse-variance weighted (IVW) fixed effect meta-analysis) of the POEM and the PIVUS results. Also in the validation step, a FDR<0.05 was used to assess significance.

STATA 16.1 (Stata inc, College Station, TX, USA) was used for these analyses.

## Results

Basic characteristics of the three samples are provided in **Table 1**.

### Metabolomics by liquid chromatography–mass spectrometry methods

In the discovery step, 193 out of the 791 evaluated metabolites were associated with WHRadjfatmass in either men or women at FDR<0.05. 154 of the metabolites were associated with fat mass.

In the replication sample of the PIVUS and the POEM cohorts, 52 of the 193 metabolites were associated with WHRadjfatmass in either men or women at FDR<0.05. Nine of those metabolites were associated with WHRadjfatmass in both men and women (**Fig 1**).

The nine validated metabolites being significantly associated with WHRadjfatmass in both sexes include ceramides, sphingomyelins and glycerophosphatidylcholines (GPCs) (see **Fig 1 and S1 Table** for details) and were all inversely related to WHRadjfatmass. Of those nine metabolites, two sphingomyelins ((d18:2/24:1, d18:1/24:2) and (d18:2/24:2)) were very far from being significantly associated with fat mass (p = 0.53 and p = 0.67, respectively), while the other 7 showed significant negative association with fat mass. For sphingomyelin (d18:1/22:2, d18:2/22:1, d16:1/24:2), the association was in opposite direction for fat mass compared to WHRadjfatmass.

Nineteen of the 52 replicated metabolites were significantly associated with WHRadjfatmass in women only (**Table 2**). Of those, six showed an interaction with sex with p<0.05. Those 6 represents different chemical classes (GPCs, fatty acids, carotenes, and bile acids). All of these 6 metabolites were also associated with fat mass.

### Lipoprotein particles by NMR

In the discovery step, 82 of the 91 evaluated metabolites were associated with WHRadjfatmass in either men or women at FDR<0.05. All of these 82 metabolites except one, very small VLDL cholesterol, were also associated with fat mass.

In the replication step, 43 of these 82 metabolites were associated with WHRadjfatmass in either men or women at FDR<0.05. Fourteen of those metabolites were related to WHR in both men and women (see **Fig 2 and S2 Table**).

**Table 1. Basic characteristics of the three samples.** Means and standard deviations (in parenthesis) or proportions are given.

|  | EpiHealth | POEM | PIVUS |
|---|---|---|---|
| **n** | 2342 | 502 | 604 |
| **Age (years)** | 61 (8.4) | 50 (0.1) | 80 (0.2) |
| **Female sex (%)** | 50% | 50% | 50% |
| **BMI (kg/m$^2$)** | 26.5 (3.8) | 26.4 (4.2) | 26.9 (4.6) |
| **Weight (%)** |  |  |  |
| Normal-weight | 37 | 41 | 35 |
| Overweight | 47 | 41 | 44 |
| Obese | 16 | 18 | 21 |
| **Waist/hip ratio (WHR)** |  |  |  |
| Total | 0.90 (0.08) | 0.90 (0.08) | 0.90 (0.07) |
| Men | 0.95 (0.06) | 0.94 (0.05) | 0.94 (0.06) |
| Women | 0.85 (0.07) | 0.87 (0.08) | 0.86 (0.06) |
| **Fat mass (%)** |  |  |  |
| Total | 30 (8.0) | 28 (8.0) | 32 (8.0) |
| Men | 24 (5.2) | 22 (5.2) | 27 (6.2) |
| Women | 36 (6.5) | 33 (7.2) | 37 (7.5) |
| **Alcohol intake** | 2.43 (2.92) (drinks/week) | NA | NA |
| **Exercise habits** | 2.29 (.8) (On a 5-grade scale) | 2.8 (1.01) (On a 4-grade scale) | 1.21 (1.31) (On a 4-grade scale) |
| **Education (%)** |  |  |  |
| <10 years | 21 | 8 | 56 |
| 10–12 years | 29 | 44 | 19 |
| >12 years | 50 | 48 | 25 |
| **Smoking** | 6.7 years of smoking | 9.8% | 3.2% |
| **Statin treatment (%)** | 10 | 3.4 | 30 |
| **Total cholesterol (mmol/l)** | 5.9 (1.1) | 5.3 (1.0) | 5.1 (1.0) |
| **Total triglycerides (mmol/l)** | 1.3 (0.8) | 1.2 (0.9) | 1.2 (0.6) |
| **LDL-cholesterol (mmol/l)** | 3.9 (1.0) | 3.4 (0.9) | 3.3 (0.9) |
| **HDL-cholesterol (mmol/l)** | 1.5 (0.4) | 1.4 (0.4) | 1.4 (0.4) |

NA = Not assessed

The estimates are from the validation meta-analysis of the PIVUS and POEM samples. The betas are for one SD change in WHR or fat mass. Also 95%CIs are given.

All of these metabolites were also related to fat mass. The betas and 95%CIs are given in S2 Table.

The 14 validated metabolites being associated with WHRadjfatmass in both sexes include different lipid fractions in very-large or large HDL particles, all being inversely related to WHRadjfatmass. In addition, very-large or large VLDL triglycerides were positively associated with WHRadjfatmass. All 14 validated metabolites were also associated with fat mass.

Only one validated metabolite was significantly associated with WHRadjfatmass in women only (Very-large HDL triglycerides, beta -2.4, SE 0.62, p-value = 8.5e10-5, with p = 0.36 in men, sex-interaction term p = 0.17).

Twenty-eight validated metabolites were associated with WHRadjfatmass in men only. They represent mainly large to small VLDL particles and medium HDL (inverse relationships vs WHRadjfatmass). None of those showed a sex-interaction at p<0.05, but all 28 metabolites were associated with fat mass (**Table 4**).

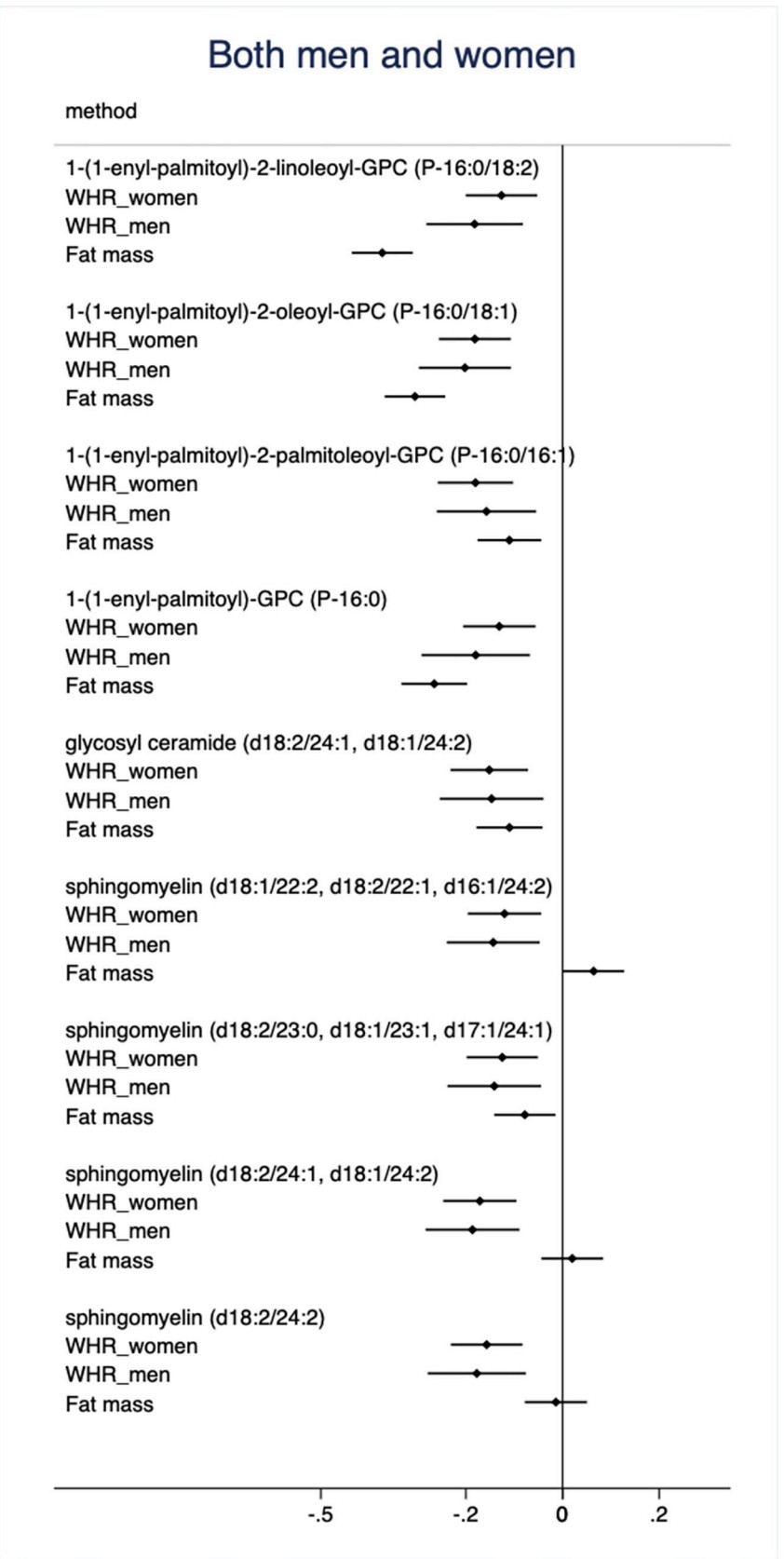

**Fig 1. Relationships between mass spectrometry-based metabolites and waist-hip ratio (WHR) in males and females and vs fat mass in both sexes combined.** Only metabolites showing a false discovery rate (FDR) <0.05 in both the discovery and validation analyses in both sexes are shown. The estimates are from the validation meta-analysis of the PIVUS and POEM samples. The betas are for one SD change in WHR or fat mass. Also 95%CIs are given.

## Discussion

The present study using a discovery/validation approach in more than 3,000 individuals showed that several metabolites were related to fat distribution independent of fat mass. Some of these associations were observed among both sexes, while a significant interaction between WHRadjfatmass and sex were seen for many metabolites. Most of the WHRadjfatmass-associated metabolites were also related to fat mass. Of particular interest were two sphingomyelins being inversely related to WHRadjfatmass in both sexes, but not associated with fat mass as such. A certain lipoprotein profile was associated with a high WHRadjfatmass, but in this case, this profile was also associated with fat mass.

### Comparison with the literature

A great number of studies have investigated the metabolic profile of obesity, as reviewed in [7]. Some studies have also evaluated the metabolomics of fat distribution [8–15], but to the best of

**Table 2. Relationships between mass spectrometry-based metabolites and waist-hip ratio adjusted for fat mass (WHRadjfatmass) in males and females and vs fat mass in both sexes combined.**

| | WHR adjusted for fat mass | | | | | | Fat mass | | |
|---|---|---|---|---|---|---|---|---|---|
| | Women | | | Men | | | | | |
| Metabolite | Beta | SE | p-value | Beta | SE | p-value | Beta | SE | p-value |
| 1,2-dipalmitoyl-GPC (16:0/16:0) | -2.57 | 0.70 | 0.0003 | -0.78 | 0.89 | 0.38 | -0.020 | 0.005 | 5.6e-05 |
| 1-(1-enyl-palmitoyl)-2-palmitoyl-GPC (P-16:0/16:0)* | -2.16 | 0.67 | 0.0012 | -2.16 | 0.91 | 0.017 | -0.034 | 0.005 | 1.6e-12 |
| 1-lignoceroyl-GPC (24:0) | -2.57 | 0.65 | 0.000073* | -1.43 | 0.90 | 0.11 | -0.042 | 0.005 | 2.1e-18 |
| 2'-O-methylcytidine | -1.94 | 0.65 | 0.0029 | -0.35 | 0.99 | 0.72 | -0.017 | 0.005 | 0.0009 |
| 3beta,7beta-dihydroxy-5-cholestenoate | 1.99 | 0.68 | 0.0034* | 0.37 | 0.88 | 0.67 | 0.033 | 0.005 | 2.0e-11 |
| branched-chain, straight-chain, or cyclopropyl 10:1 fatty acid | -2.39 | 0.66 | 0.0003* | -0.32 | 0.93 | 0.73 | -0.023 | 0.005 | 5.2e-06 |
| carotene diol (2) | -1.76 | 0.66 | 0.0073* | -0.77 | 0.86 | 0.37 | -0.040 | 0.005 | 4.1e-17 |
| chenodeoxycholic acid sulfate | 1.52 | 0.53 | 0.0042* | -0.91 | 0.81 | 0.26 | 0.015 | 0.004 | 0.00022 |
| corticosterone | -1.83 | 0.60 | 0.0023 | -1.47 | 0.87 | 0.089 | -0.016 | 0.005 | 0.00072 |
| lactosyl-N-nervonoyl-sphingosine (d18:1/24:1)* | -2.32 | 0.70 | 0.0010 | -1.58 | 0.92 | 0.084 | -0.021 | 0.005 | 2.1e-05 |
| metabolonic lactone sulfate | 2.05 | 0.62 | 0.0010* | 1.56 | 0.85 | 0.066 | 0.059 | 0.005 | 9.6e-37 |
| palmitoyl sphingomyelin (d18:1/16:0) | -1.86 | 0.65 | 0.0041 | -1.92 | 0.91 | 0.036 | -0.030 | 0.005 | 8.0e-10 |
| sphingomyelin (d17:1/14:0, d16:1/15:0)* | -2.01 | 0.64 | 0.0018 | -1.43 | 0.84 | 0.090 | -0.006 | 0.005 | 0.24 |
| sphingomyelin (d17:1/16:0, d18:1/15:0, d16:1/17:0)* | -2.19 | 0.65 | 0.0007 | -1.69 | 0.89 | 0.057 | -0.016 | 0.005 | 0.00098 |
| sphingomyelin (d17:2/16:0, d18:2/15:0)* | -1.70 | 0.59 | 0.0043 | -1.36 | 0.78 | 0.084 | 0.028 | 0.004 | 1.3e-10 |
| sphingomyelin (d18:1/24:1, d18:2/24:0)* | -2.24 | 0.65 | 0.0006 | -1.16 | 0.92 | 0.21 | -0.002 | 0.005 | 0.74 |
| sphingomyelin (d18:2/14:0, d18:1/14:1)* | -2.09 | 0.57 | 0.0002 | -1.71 | 0.72 | 0.017 | 0.035 | 0.004 | 1.1e-17 |
| sphingomyelin (d18:2/16:0, d18:1/16:1)* | -1.84 | 0.61 | 0.0026 | -1.41 | 0.84 | 0.095 | 0.026 | 0.005 | 1.5e-08 |
| sphingomyelin (d18:2/23:1)* | -2.09 | 0.62 | 0.0007 | -1.99 | 0.77 | 0.0099 | 0.007 | 0.004 | 0.096 |

Only metabolites showing a false discovery rate (FDR) p-value <0.05 in both the discovery and validation analyses in women are shown. The beta, SE and p-values are from the validation meta-analysis of the PIVUS and POEM samples

* following the p-value in females denotes an interaction with sex with p<0.05. The table is sorted by the p-value for WHR adjusted for fat mass in women.

Twenty-four validated metabolites were associated with WHR in men only (**Table 3**). Of those, five showed an interaction with sex with p<0.05 (one GPE and four amino acid derivatives). All of these 5 metabolites were also related to fat mass.

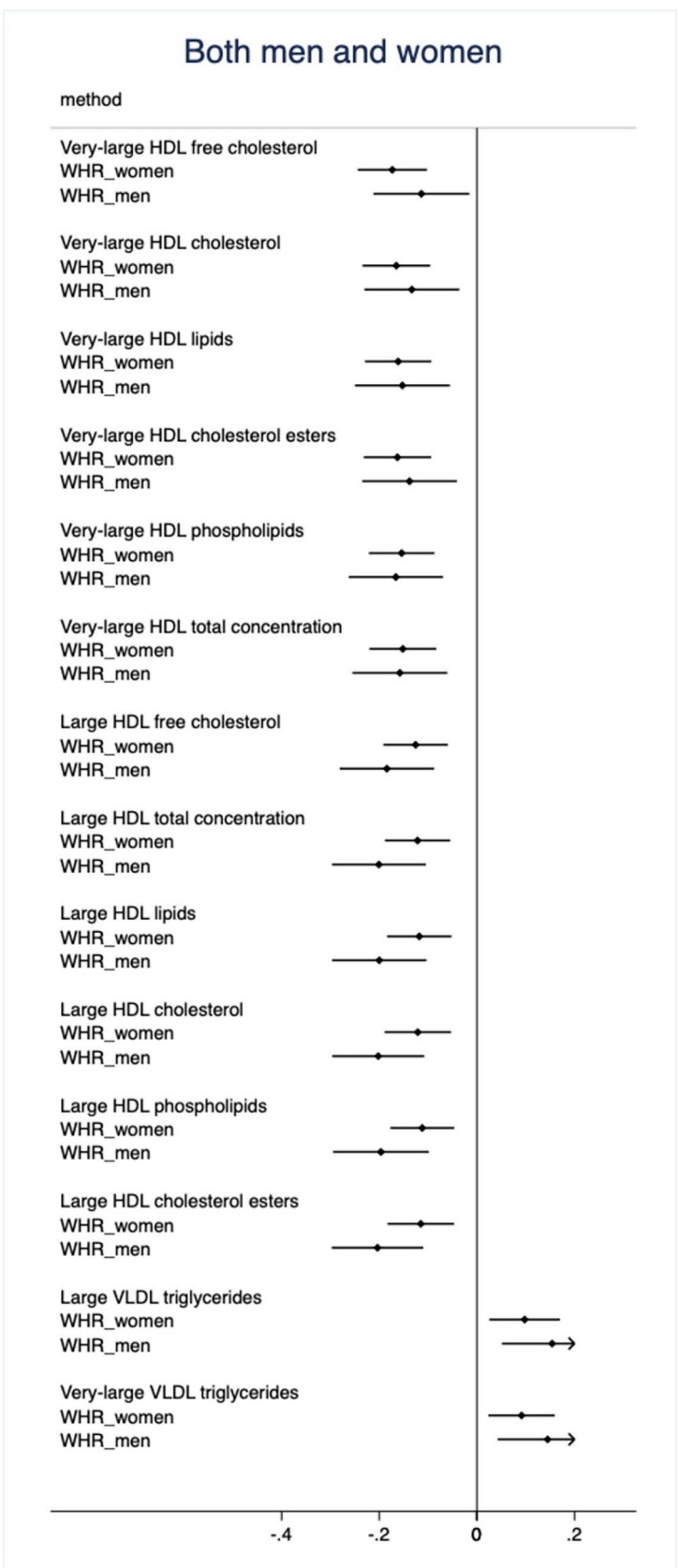

**Fig 2. Relationships between nuclear magnetic resonance spectrometry-based lipoprotein metabolites and waist-hip ratio (WHR) in males and females in both sexes combined.** Only metabolites showing a false discovery rate (FDR) <0.05 in both the discovery and validation analyses in both sexes are shown.

our knowledge, no other study have tried to link metabolites to WHR when taking fat mass into account. This approach aimed to identify metabolites linked to an altered fat distribution, independently of general obesity.

The study from the EPIC-Potsdam cohort [14], including Germans with almost exclusively European descent, studied the waist circumference (adjusted for hip circumference) and hip circumference (adjusted for waist circumference) and came to the conclusion that the metabolic profile for waist circumference was similar to that of BMI, but hip circumference showed a unique metabolomic profile. That study did however not evaluate WHR, but they showed isoleucine and several phosphatidylcholines (aa or ae C42:0, C34:3, C42:4, C42:5, C44:4 and C44:6) to show different directions in their relationships with waist and hip circumference. None of these metabolites were however related to WHRajdfatmass in our study.

**Table 3. Relationships between mass spectrometry-based metabolites and waist-hip ratio (WHR) in males and females and vs fat mass in both sexes combined.**

| | WHR adjusted for fat mass | | | | | | Fat mass | | |
|---|---|---|---|---|---|---|---|---|---|
| | Women | | | Men | | | | | |
| Metabolite | Beta | SE | p-value | Beta | SE | p-value | Beta | SE | p-value |
| 1-carboxyethylisoleucine | 0.84 | 0.57 | 0.14 | 3.26 | 0.86 | 0.0001 | 0.026 | 0.0045 | 4.1e-09 |
| 1-carboxyethylleucine | 0.84 | 0.59 | 0.15 | 2.98 | 0.88 | 0.0007 | 0.033 | 0.0046 | 2.8e-13 |
| 1-carboxyethylphenylalanine | 0.91 | 0.67 | 0.17 | 3.39 | 0.87 | 9.2e-05 | 0.029 | 0.0048 | 1.3e-09 |
| 1-carboxyethyltyrosine | 0.23 | 0.47 | 0.62* | 2.79 | 0.83 | 0.0008 | 0.027 | 0.0041 | 9.2e-11 |
| 1-carboxyethylvaline | 0.75 | 0.67 | 0.27 | 3.17 | 0.88 | 0.0003 | 0.029 | 0.0048 | 8.7e-10 |
| 1-palmitoyl-2-oleoyl-GPE (16:0/18:1) | 0.42 | 0.68 | 0.54 | 2.85 | 0.93 | 0.0021 | 0.025 | 0.0050 | 4.8e-07 |
| 1-stearoyl-2-oleoyl-GPE (18:0/18:1) | 0.11 | 0.69 | 0.87* | 2.79 | 0.95 | 0.0032 | 0.027 | .0051 | 9.8e-08 |
| 2-O-methylascorbic acid | 0.052 | 0.65 | 0.94 | 2.59 | 0.95 | 0.0063 | 0.041 | .0049 | 7.1e-17 |
| 2-aminoadipate | 0.90 | 0.63 | 0.15* | 2.66 | 0.84 | 0.0016 | 0.030 | 0.0046 | 9.0e-11 |
| 2-aminoheptanoate | -0.44 | 0.72 | 0.54 | 2.68 | 0.91 | 0.0033 | 0.013 | 0.0051 | 0.01 |
| 4-hydroxyglutamate | 1.35 | 0.56 | 0.02 | 3.29 | 0.90 | 0.0002 | 0.041 | 0.0045 | 4.8e-20 |
| N-acetylglycine | -0.34 | 0.67 | 0.61 | -2.48 | 0.88 | 0.0047 | -0.027 | 0.0048 | 2.3e-08 |
| N-acetylkynurenine | 1.29 | 0.55 | 0.02* | 2.67 | 0.89 | 0.0026 | 0.017 | 0.0044 | 0.00012 |
| N-acetylphenylalanine | 0.71 | 0.66 | 0.29 | 2.41 | 0.91 | 0.0079 | 0.027 | 0.0049 | 2.0e-08 |
| aspartate | 0.51 | 0.67 | 0.44* | 2.78 | 0.89 | 0.0017 | 0.030 | 0.0048 | 6.2e-10 |
| beta-citrylglutamate | 1.17 | 0.68 | 0.085 | 2.77 | 0.96 | 0.0038 | 0.020 | 0.0051 | 0.0001 |
| cortolone glucuronide | 1.32 | 0.60 | 0.027 | 2.32 | 0.87 | 0.0077 | 0.064 | 0.0046 | 0 |
| gamma-glutamylglutamate | 0.92 | 0.62 | 0.13 | 3.10 | 0.91 | 0.0006 | 0.035 | 0.0047 | 9.7e-14 |
| gamma-glutamylglutamine | -1.17 | 0.69 | 0.090 | -2.71 | 0.95 | 0.0046 | -0.013 | 0.0051 | 0.008 |
| gamma-glutamylisoleucine | 0.61 | 0.57 | 0.28 | 2.58 | 0.81 | 0.0013 | 0.042 | 0.0043 | 1.1e-22 |
| glutamate | 1.50 | 0.65 | 0.021 | 3.38 | 0.87 | 0.0001 | 0.035 | 0.0048 | 4.6e-13 |
| hydroxy-CMPF | -0.94 | 0.65 | 0.15 | -2.64 | 0.89 | 0.0031 | -0.024 | 0.0048 | 4.8e-07 |
| sphingomyelin (d18:1/20:1, d18:2/20:0) | -1.40 | 0.64 | 0.029 | -2.79 | 0.85 | 0.0010 | 0.018 | 0.0047 | 9.1e-05 |
| urate | 1.53 | 0.57 | 0.0073 | 2.56 | 0.79 | 0.0012 | 0.041 | 0.0043 | 4.7e-21 |

Only metabolites showing a false discovery rate (FDR) <0.05 in both the discovery and validation analyses in men are shown. The beta, SE and p-values are from the validation meta-analysis of the PIVUS and POEM samples

*following the p-value in females denotes an interaction with sex with p<0.05. The table is sorted by the p-value for WHR adjusted for fat mass in men.

**Table 4. Relationships between nuclear magnetic resonance spectrometry-based lipoprotein metabolites and waist-hip ratio (WHR) in males and females and vs fat mass in both sexes combined.**

| Lipoprotein | WHR adjusted for fat mass | | | | | | Fat mass | | |
|---|---|---|---|---|---|---|---|---|---|
| | Women | | | Men | | | | | |
| | Beta | SE | p-value | Beta | SE | p-value | Beta | SE | p-value |
| Medium HDL cholesterol esters | -1.18 | 0.61 | 0.054 | -3.19 | 0.85 | 0.0002 | -0.028 | 0.0046 | 1.2e-09 |
| Medium HDL cholesterol | -1.17 | 0.61 | 0.054 | -3.11 | 0.85 | 0.0002 | -0.028 | 0.0046 | 2.4e-09 |
| Small VLDL triglycerides | 1.52 | 0.67 | 0.024 | 2.74 | 0.85 | 0.0013 | 0.042 | 0.0048 | 4.1e-18 |
| Large VLDL phospholipids | 1.59 | 0.63 | 0.011 | 2.85 | 0.89 | 0.0014 | 0.040 | 0.0047 | 1.8e-17 |
| Small VLDL total concentration | 1.12 | 0.67 | 0.097 | 2.70 | 0.86 | 0.0017 | 0.041 | 0.0048 | 3.3e-17 |
| Small HDL triglycerides | 1.44 | 0.68 | 0.034 | 2.64 | 0.85 | 0.0019 | 0.043 | 0.0048 | 1.8e-19 |
| Medium HDL free cholesterol | -1.32 | 0.59 | 0.025 | -2.69 | 0.87 | 0.0021 | -0.023 | 0.0045 | 4.3e-07 |
| Medium VLDL triglycerides | 1.53 | 0.64 | 0.017 | 2.67 | 0.87 | 0.0022 | 0.041 | 0.0047 | 6.1e-18 |
| Small VLDL lipids | 0.91 | 0.66 | 0.170 | 2.60 | 0.85 | 0.0023 | 0.039 | 0.0048 | 2.4e-16 |
| Large VLDL total concentration | 1.54 | 0.62 | 0.013 | 2.62 | 0.88 | 0.0027 | 0.041 | 0.0047 | 3.2e-18 |
| Large VLDL lipids | 1.51 | 0.62 | 0.015 | 2.62 | 0.88 | 0.0028 | 0.041 | 0.0047 | 3.8e-18 |
| Very-large VLDL phospholipids | 1.31 | 0.59 | 0.026 | 2.61 | 0.88 | 0.0029 | 0.039 | 0.0046 | 1.3e-17 |
| Large VLDL free cholesterol | 1.50 | 0.63 | 0.018 | 2.55 | 0.86 | 0.0030 | 0.039 | 0.0047 | 2.4e-17 |
| Medium HDL total concentration | -1.16 | 0.60 | 0.052 | -2.65 | 0.89 | 0.0030 | -0.022 | 0.0046 | 2.8e-06 |
| Small VLDL phospholipids | 0.56 | 0.65 | 0.40 | 2.53 | 0.89 | 0.0043 | 0.035 | 0.0048 | 4.3e-13 |
| Large VLDL cholesterol | 1.19 | 0.63 | 0.061 | 2.47 | 0.87 | 0.0047 | 0.039 | 0.0047 | 3.0e-16 |
| Very-large VLDL lipids | 1.38 | 0.58 | 0.018 | 2.45 | 0.87 | 0.0048 | 0.040 | 0.0046 | 3.3e-18 |
| Very-large VLDL total concentration | 1.43 | 0.58 | 0.014 | 2.43 | 0.87 | 0.0052 | 0.039 | 0.0046 | 6.3e-18 |
| Large VLDL cholesterol esters | 1.01 | 0.63 | 0.110 | 2.42 | 0.88 | 0.0059 | 0.037 | 0.0048 | 5.2e-15 |
| Very-large VLDL free cholesterol | 1.34 | 0.59 | 0.024 | 2.36 | 0.86 | 0.0062 | 0.038 | 0.0046 | 2.8e-17 |
| Small VLDL cholesterol esters | 0.49 | 0.66 | 0.460 | 2.27 | 0.83 | 0.0064 | 0.037 | 0.0047 | 6.6e-15 |
| Very-large VLDL cholesterol | 1.12 | 0.60 | 0.060 | 2.33 | 0.86 | 0.0070 | 0.038 | 0.0046 | 1.4e-16 |
| Medium HDL lipids | -1.09 | 0.60 | 0.067 | -2.43 | 0.90 | 0.0071 | -0.016 | 0.0046 | 4.1e-04 |
| Very-large VLDL cholesterol esters | 0.99 | 0.60 | 0.10 | 2.30 | 0.86 | 0.0077 | 0.037 | 0.0046 | 1.6e-15 |
| Very small VLDL triglycerides | 1.06 | 0.69 | 0.13 | 2.38 | 0.93 | 0.010 | 0.037 | 0.0049 | 3.8e-14 |
| Small VLDL cholesterol | 0.42 | 0.65 | 0.52 | 2.13 | 0.84 | 0.011 | 0.035 | 0.0047 | 9.6e-14 |
| Medium VLDL total concentration | 0.67 | 0.64 | 0.30 | 2.08 | 0.89 | 0.019 | 0.034 | 0.0047 | 1.4e-12 |
| Medium VLDL phospholipids | 0.49 | 0.64 | 0.44 | 2.12 | 0.90 | 0.019 | 0.030 | 0.0048 | 4.2e-10 |

Only metabolites showing a false discovery rate (FDR) <0.05 in men only are shown. The beta, SE and p-values are from the validation meta-analysis of the PIVUS and POEM samples. The table is sorted by the p-value for WHR adjusted for fat mass in men.

In the other study based on three other Swedish samples using around 200 named metabolites [15], with almost exclusively European descent, one sphingomyelin (32:2) were amongst the metabolites found to be related to WHRadjBMI. In that study fat mass was not evaluated directly. In the present study, we found several sphingomyelin with a higher number of carbons to be linked to WHRadjfatmass. Neither could we find any other association being similar across the two studies. If that was due to the use of adjustment for fat mass in one study and BMI in the other is not known.

We identified two sphingomyelins of particular interest ((d18:2/24:1, d18:1/24:2) and (d18:2/24:2)) being inversely related to central adiposity in both sexes, but not being linked to fat mass. It is not likely that we would not be able to detect a relevant relationship between a metabolite and fat mass, since we have an 80% power to detect a relationship with an $R^2$ of 0.0071, and for those two sphingomyelins the relationships vs fat mass were far from being significant.

Sphingomyelin (SM) is a class of sphingolipids formed by adding phosphocholine to ceramide. As recently reviewed in [26], sphingolipids were initially thought to merely be structural components of the cell membrane, but recently a number of regulatory properties have been coupled to sphingolipids, such as cell growth and death, inflammation, angiogenesis and metabolism. Low levels of SMs have also recently been associated with cardiovascular diseases, such as stroke [27] and heart failure [28], as well as with the structure of the arterial wall [29]. Thus, certain sphingolipids might be a link between an altered fat distribution and cardiovascular disease. Unfortunately, no GWAS studies for these sphingomyelins with a 42-carbon chain length are available according to GWAS catalogue (https://www.ebi.ac.uk/gwas/) and the Helmholtz/KORA Metabolite GWAS server (http://metabolomics.helmholtz-muenchen.de/pgwas/), so Mendelian randomization studies cannot be carried out in order to investigate causality.

Why some of the SM were related to WHR and not others, and why two SMs were related to WHR and not fat mass, while other were related to both WHR and fat mass is not known. One explanation for this finding is that only slight changes in the SM molecule could have profound effects on the physical properties. This is exemplified in a paper on lipidomics and obesity conducted in Sweden with almost exclusively European descent, in which it was found that SM 34:1;2 has the greatest negative and SM 34:2;2 the greatest positive estimates vs fat mass [10]. Thus, only a change in a double bond could result in different signs of the association vs fat mass. In another study, high levels of serum SM species with distinct saturated acyl chains (C18:0, C20:0, C22:0 and C24:0) closely correlate with the parameters of obesity in young adult Japanese individuals [30], but this is not a universal finding, as could be seen in a study sample collected in Iran [31]. Thus, it is likely that all SM should not be regarded as equal, although we do not have the knowledge today to understand how the chemical properties of different SMs translates to relationships vs fat mass and fat distribution.

Another question arises if SMs are causally related to fat mass and fat distribution. An experimental study using knock-out of the enzyme involved in SM synthesis (SMS2) in mice supports a role of SMs in obesity [32]. Mice with SMS2 deficiency developed obesity when challenged with a high-fat diet. However, also the other direction of the relationship between SMs and fat mass might exist. In a study of weight loss induced by a low-caloric diet conducted in Spain and Denmark, the change in fat mass over 8 weeks was inversely related to the change in certain SMs (33:1, 35:1, 36:0, and 36:1), while the change in fat mass was directly related to the change in other SMs (32:1, 32:2, 38:1, 40:1 and 41:1) [33].

Apart from the 9 metabolites linked to WHRadjfatmass in both sexes, we identified a number of metabolites being related to WHRadjfatmass in one sex only. This is not surprising since WHR is very different in men and women and that most genetic correlates to WHR also are sex-specific. Both the male-specific and the female-specific metabolites being related to WHR comes from several chemical classes and it is hard to see any clear pattern in these two metabolomic profiles. Again, genetic studies might be a way forward to disentangle these sex-specific metabolomic profiles in the future.

As with the MS-based metabolomics analysis, the analysis of the lipoprotein profile showed several validated metabolites to be linked to a high WHRadjfatmass. It was mainly very-large and large HDL-cholesterol (inverse) and large VLDL-triglycerides that were related to WHRadjfatmass in both sexes. In the sex-stratified analyses, associations with WHRadjfatmass were much more common in males than in females, with predominantly small VLDL being sex-specific. Very-large and large HDL-cholesterol (inverse) and large VLDL-triglycerides have been linked to cardiovascular disease [16], but these lipoprotein fractions were also related to fat mass, not only to fat distribution.

In the relationship between WHRadjfatmass and metabolites, it is likely that most relationships have a causal direction from an unfavorable fat distribution to a change in metabolites rather than the opposite, since a major weight change induces profound metabolic effects [33, 34]. Since we also know that WHR is a major risk factor for CVD both in observational [35] and genetic studies [3], it would be of interest to find metabolites being mediators in the causal WHR->CVD relationship. Regarding the 14 lipoprotein measurements identified (Fig 2) to be linked to WHRadjfatmass in both men and women, a mediating role for those lipoprotein-based metabolites are plausible, since they all have been linked to atherosclerosis and incident CVD [16] with the same direction of associations as found vs WHRadjfatmass in the present study. As discussed above, the role of the 9 MS-based metabolites (Table 1) are less clear, since the knowledge on sphingomyelins and GPCs in CVD are much less advanced than the knowledge on lipoproteins in CVD. We are still awaiting to obtain robust, powerful genetic instruments for different sphingomyelins and GPCs that are not pleotropic to be used in Mendelian randomization studies. Also mice models with knock-out of different sphingomyelins and GPCs on a genetic atherosclerotic background would be a way forward to disentangle if sphingomyelins and GPCs are important players in atherosclerosis formation.

It is also be emphasized that the MS-based part of the study should be regarded as an untargeted approach, since a large number of metabolites from a great number of chemical classes were analyzed, and therefore the evaluation of the WHRadjfatmass vs metabolite associations were hypothesis-free, like in a GWAS study. It is therefore not surprising that many of the findings were not expected and given the unperfect knowledge of many of the metabolites, the findings are hard to understand in detail at this stage.

The strength of the present study is the use of the same MS-based metabolomic platform and lipoprotein NMR analysis with a large number of metabolites and lipoproteins subfractions in three different studies, so we could obtain validated relationships.

One limitation is the cross-sectional nature of the studies, which hinders any conclusion of causal directions. Another limitation is the fact that we do not have genetic instruments for the metabolites of interest to be used in Mendelian randomization studies. Metabolomic measurements were performed by a commercial company. Some of the details regarding standards and QC determinations were considered proprietary to their platform and therefore not shared. How this translates into the findings in the present study is unknown, but as a general rule any poor performance of a technique would only increase the probability of the null hypothesis and would not produce any false positive findings.

We used bioimpedance to evaluate fat mass in the present study, a technique that has been present for decades. However, DEXA is the gold standard in this respect, but was only used in one of the samples, the POEM study, being the smallest of the three cohorts used. Therefore, we did not use DEXA due to the limited power and lack of replication sample. However, when we related bioimpedance to DEXA measurements for fat mass, the correlation was very good (correlation coefficient 0.93), so we regard the bioimpedance measurement of fat mass to be valid, in accordance with previous evaluations of the bioimpedance technique [22, 23].

DEXA has the advantage over bioimpedance that regional fat distribution could be assessed in detail. In the absence of DEXA, we used an indirect measurement of regional fat distribution, the WHR, a measure that is a better predictor of myocardial infarction than measurements of general obesity, such as BMI [35]. WHR has the advantage over DEXA of being a cheap and easy measurement and is widely used in the clinic, but it would be of great interest to validate our present findings in a sample with DEXA measurements.

The samples used in the present studies are almost exclusively including subjects with a European descent. In order to generalize the findings to other populations, replication studies in subjects from other parts of the world has to be undertaken, especially since most other

studies in this field also have been conducted in subjects with a European descent. It is also emphasized that the MS-based part of the study should be regarded as an untargeted approach, since a large number of metabolites from a great number of chemical classes were analyzed, and therefore the evaluation of the WHRadjfatmass vs metabolite associations were hypothesis-free, like in a GWAS study. It is therefore not surprising that many of the findings were not expected and given the imperfect knowledge of many of the metabolites, the findings are hard to understand in detail at this stage. Regarding the lipoprotein-based metabolites, they were preselected to cover the lipoprotein spectra and therefore this analysis should be considered as targeted and due to the greater knowledge on lipoproteins, these results could be interpreted more in detail.

Thyroid function might have been a confounder in the present evaluation of the metabolomic profile of WHR, but unfortunately we do not have valid measurements of thyroid function in the samples.

In conclusion, two sphingomyelins were inversely linked to WHR (fat mass adjusted) in both men and women without being related to fat mass, while very-large or large HDL particles were inversely related to WHR as well as to fat mass. If these sphingomyelins represent a link between WHR and cardiometabolic diseases remains to be established when genetic instruments might become available in the future.

## Supporting information

**S1 Table. Relationships between mass spectrometry-based metabolites and waist-hip ratio (WHR) in males and females and vs fat mass in both sexes combined.** Only metabolites showing a false discovery rate (FDR) <0.05 in both the discovery and validation analyses in both sexes are shown. The beta, SE and p-values are from the validation meta-analysis of the PIVUS and POEM samples.
(XLSX)

**S2 Table. Relationships between nuclear magnetic resonance spectrometry-based lipoprotein metabolites and waist-hip ratio (WHR) in males and females and vs fat mass in both sexes combined.** Only metabolites showing a false discovery rate (FDR) <0.05 in both the discovery and validation analyses in both sexes are shown. The beta, SE and p-values are from the validation meta-analysis of the PIVUS and POEM samples.
(XLSX)

## Author Contributions

**Conceptualization:** Lars Lind, Tove Fall.

**Formal analysis:** Lars Lind.

**Investigation:** Lars Lind, Shafqat Ahmad, Sölve Elmståhl, Tove Fall.

**Writing – original draft:** Lars Lind.

**Writing – review & editing:** Shafqat Ahmad, Sölve Elmståhl, Tove Fall.

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
