## [Decision Letter · Decision Letter 0]

28 Nov 2022

PONE-D-22-29652THE METABOLIC PROFILE OF WAIST TO HIP RATIO – A MULTI-COHORT STUDYPLOS ONE

Dear Dr. Lind,

Thank you for submitting your manuscript to PLOS ONE. After careful consideration, we feel that it has merit but does not fully meet PLOS ONE’s publication criteria as it currently stands. Therefore, we invite you to submit a revised version of the manuscript that addresses the points raised during the review process. Please response the comments from two reviewers and revise the manuscript accordingly.

We look forward to receiving your revised manuscript.

Kind regards,

Xu Chen, Ph.D.

Academic Editor

PLOS ONE

Journal Requirements:

Additional Editor Comments:

This study investigated the associations of 793 metabolites, 97 lipoproteins with waist/hip ratio adjusted for fat mass and fat mass, in three cohorts (1 discovery and 2 replications).

In general, the work is enormous and statistical analyses are acceptable. Please response the comments from the reviewers and revise the manuscript accordingly.

Reviewers' comments:

Reviewer's Responses to Questions

**Comments to the Author**

1. Is the manuscript technically sound, and do the data support the conclusions?

Reviewer #1: Yes

Reviewer #2: Partly

2. Has the statistical analysis been performed appropriately and rigorously? 

Reviewer #1: Yes

Reviewer #2: N/A

3. Have the authors made all data underlying the findings in their manuscript fully available?

Reviewer #1: No

Reviewer #2: No

4. Is the manuscript presented in an intelligible fashion and written in standard English?

Reviewer #1: Yes

Reviewer #2: Yes

5. Review Comments to the Author

Reviewer #1: This is a well conceived, executed, and written study describing associations between metabolites and fat distribution. I am not a statistician, but the statistical analysis and design seem quite rigorous, and the results partially reflect expectations. The unexpected results are difficult to make biological sense of, but this, as the authors point out, is one of the benefits of untargeted omics. Some specific comments:

the details of the analytical platforms are rather thin, though they are well established.

inverse-rank transformed - is this the same as inverse-normal? https://cran.r-project.org/web/packages/RNOmni/vignettes/RNOmni.html

carotens - should this be carotenes?

"structural components of the cell wall" - should be "structural components of the cell membrane" i think.

"The samples used in the present studies are almost exclusively including subjects with a European descent." shouldn't this also be taken into account earlier in the discussion when trying to compare the results of this study with other literature?

"The data could be shared with other researchers after a request to the steering committee (karl.michaelsson@surgsci.uu.se)" - the contact information for this request would ideally not be tied to a single person. a general account (i.e. data.steering.committee@surgsci.uu.se) can be created that is accessible to more than one individual. Small issue, perhaps, but this is a great set of data and maximizing its availability within the assigned bounds seems worthwhile.

Reviewer #2: The manuscriped described a cross-sectional study to investigate the metabolic profile of waist to hip ratio, generally the manuscript is interesting and well organized, however, some issues should be addressed:

1. please provid the ethical approval number;

2. The authors claimed that 793 metabolites were determined (abstract), however, in the method part 791 metabolites in line 159, page 7.

3. LC-MS rather than High resolution MS was used, how authors separate possible interference with similar m/s AND retention time?

4. In hte lcms method, 791/793? molecules were monitored, this was not untarget metabolics, this should be target metabolics.

4. line 161, page 7, please provide more information of the library, like company, country, et al.

5. Please provide instrument information of LC-MS used, as well as columns used.

6. In LC-MS methods, internal standard for each sample is appreciated, the authors just claimed that several IS used(Line 149), how many IS the authors used? and please declear why they choose these IS?

7. More information should be provide for metabolomics, the volumn of plasma used, the methanol volume used?

8. How the authors quantify the 791or 793 molecules without calibrators?

9. Authors should provide the CV for the quality controls to monitor the bias when determining samples in different batches.

6. PLOS authors have the option to publish the peer review history of their article (what does this mean?). If published, this will include your full peer review and any attached files.

Reviewer #1: **Yes: **Corey D Broeckling

Reviewer #2: No

---

## [Author Response · Author response to Decision Letter 0]

7 Feb 2023

Reviewer #1: This is a well conceived, executed, and written study describing associations between metabolites and fat distribution. I am not a statistician, but the statistical analysis and design seem quite rigorous, and the results partially reflect expectations. The unexpected results are difficult to make biological sense of, but this, as the authors point out, is one of the benefits of untargeted omics. Some specific comments:

the details of the analytical platforms are rather thin, though they are well established.

Reply: This part has now been expanded.

inverse-rank transformed - is this the same as inverse-normal? https://cran.r-project.org/web/packages/RNOmni/vignettes/RNOmni.html

Reply: Yes, you are right. It appears to have different versions of the name. We have now changed the wording to: rank based inverse normal transformation (line 237). We have used this transformation in several papers om metabolomics or proteomics, such as: 

1. Lind L, Fall T, Ärnlöv J, Elmståhl S, Sundström J. Large-Scale Metabolomics and the Incidence of Cardiovascular Disease. J Am Heart Assoc. 2023;12(2):e026885.

2. Lind L, Sundström J, Elmståhl S, Dekkers KF, Smith JG, Engström G, et al. The metabolomic profile associated with clustering of cardiovascular risk factors-A multi-sample evaluation. PLoS One. 2022;17(9):e0274701.

3. Lind L, Zanetti D, Ingelsson M, Gustafsson S, Ärnlöv J, Assimes TL. Large-Scale Plasma Protein Profiling of Incident Myocardial Infarction, Ischemic Stroke, and Heart Failure. J Am Heart Assoc. 2021;10(23):e023330.

carotens - should this be carotenes?

Reply: Yes, you are right. This typo has now been corrected (line 292).

"structural components of the cell wall" - should be "structural components of the cell membrane" i think.

Reply: Yes, you are right again. We have now changed wall to membrane (line 381).

"The samples used in the present studies are almost exclusively including subjects with a European descent." shouldn't this also be taken into account earlier in the discussion when trying to compare the results of this study with other literature?

Reply: We have now looked through the papers being cited in the discussion on metabolomics and obesity/WHR and commented upon the location and ethnicity of the studied sample. It turned out that these other studies most often were carried out in subjects with a European descent.

We have therefore also added to the discussion part (line 475-478): The samples used in the present studies are almost exclusively including subjects with a European descent. In order to generalize the findings to other populations, replication studies in subjects from other parts of the world has to be undertaken, especially since most other studies in this field also have been conducted in subjects with a European descent.

"The data could be shared with other researchers after a request to the steering committee (karl.michaelsson@surgsci.uu.se)" - the contact information for this request would ideally not be tied to a single person. a general account (i.e. data.steering.committee@surgsci.uu.se) can be created that is accessible to more than one individual. Small issue, perhaps, but this is a great set of data and maximizing its availability within the assigned bounds seems worthwhile.

Reply: We agree that relying on a single person is vulnerable. However, creating a new account which nobody takes the responsibility to look after is even more dangerous. We propose to use the account that goes to the administrative office of our institution, an account that is looked at every day by several administrators, who could forward any request to us working with the EpiHealth study. We believe that is the safest way to handle the problem.

Line 500-502: “The data could be shared with other researchers after a request to the steering committee, mailed to the administrative office at the department (prefekt@medsci.uu.se)”

Reviewer #2: The manuscriped described a cross-sectional study to investigate the metabolic profile of waist to hip ratio, generally the manuscript is interesting and well organized, however, some issues should be addressed:

Reply: Generally, most questions below are regarding technical details on the MS-based method. These measurements were performed by a commercial company. We have now after some time got them to answer most of the questions below and we have added their answers to the methods section. Since none of the authors of the paper is an analytical chemist, we have to rely on their comments.

1. please provid the ethical approval number;

Reply: This has now been given (line 114-115).

2. The authors claimed that 793 metabolites were determined (abstract), however, in the method part 791 metabolites in line 159, page 7.

Reply: Thanks for noticing this typo. We have now changed to 791 throughout the manuscript.

3. LC-MS rather than High resolution MS was used, how authors separate possible interference with similar m/s AND retention time?

Reply from Metabolon: Please see answer to Question 5.

4. In hte lcms method, 791/793? molecules were monitored, this was not untarget metabolics, this should be target metabolics.

Reply: We have now deleted word non-targeted (line 147).

5. line 161, page 7, please provide more information of the library, like company, country, et al.

Reply from Metabolon (line 175-185): Metabolon has built and maintains a proprietary chemical library based on authentic standards that contains the retention time (RT), retention index (RI), mass to charge ratio (m/z), and mass spectral data (including MS/MS spectral data) on all molecules present in the library per method. Biochemical identifications are therefore based on three criteria: retention index within a narrow retention window of the proposed identification, accurate mass match to the library, and the MS/MS forward and reverse scores between the experimental data and authentic standards. The MS/MS scores are based on a comparison of the ions present in the experimental spectrum to the ions present in the library spectrum. While there may be similarities between molecules based on one of these factors, the use of all three data points can be used to accurately identify biochemicals.

6. Please provide instrument information of LC-MS used, as well as columns used.

Reply from Metabolon (line 159-163): Metabolon’s untargeted metabolomics panel utilizes a Waters ACQUITY ultra-performance liquid chromatography (UPLC) and a Thermo Scientific Q-Exactive or Q-Exactive Plus high resolution/accurate mass spectrometer interfaced with a heated electrospray ionization (HESI-I). The columns utilized were Waters BEH C18 2.1X100mm 1.7µm and Waters BEH Amide 2.1 x150mm 1.7µm. 

7. In LC-MS methods, internal standard for each sample is appreciated, the authors just claimed that several IS used(Line 149), how many IS the authors used? and please declear why they choose these IS?

Reply from Metabolon (line 193-204): IS are used for alignment of data and for QC of instrument performance. The IS were selected to span the chromatogram and allow the creation of an RI ladder. Furthermore, they were selected to be representative of the type of endogenous compounds detected and therefore can be used to monitor consistency of chromatographic behavior and MS response.

Process standards are added during sample extraction to ensure consistent performance of the entire process from sample preparation through sample analysis.

Instrument Performance Standards: d35-octadecanoic acid, fluorophenylglycine, d5-indole acetate, chlorophenylalanine, Br-phenylalanine, d5-tryptophan, d4-tyrosine, d3-serine, d3-aspartic acid, d7-ornithine, d4-lysine.

Process Assessment Standards: Fluorophenylglycine, chlorophenylalanine.

8. More information should be provide for metabolomics, the volumn of plasma used, the methanol volume used?

Reply from Metabolon (line 148-149): 100ul of human plasma was utilized for analysis. 500ul of methanol was added to each sample, then mixed for 2 minutes in a GenoGrinder before centrifugation for 10minutes at 680g.

9. How the authors quantify the 791or 793 molecules without calibrators?

Reply from Metabolon (line 165-170): The relative concentration of identified peaks associated with each chemical in the Metabolon library (where present), are obtained by measuring the area of the peak relative to the surrounding baseline. All peak areas are integrated for each biochemical, based on the authentic standard for each biochemical, providing a consistent quantitation of relative abundance. 

10. Authors should provide the CV for the quality controls to monitor the bias when determining samples in different batches.

Reply from Metabolon: Metabolon does not provide the QC data for the plasma QC samples as this is considered proprietary to their platform.

Reply from authors: Since Metabolon does not provide all the requested data, we have now added to the limitation section (line 457-461): ”Metabolomic measurements were performed by a commercial company. Some of the details regarding standards and QC determinations were considered proprietary to their platform and therefore not shared. How this translates into the findings in the present study is unknown, but as a general rule any poor performance of a technique would only increase the probability of the null hypothesis and would not produce any false positive findings.”

---

## [Editor Report · Decision Letter 1]

15 Feb 2023

THE METABOLIC PROFILE OF WAIST TO HIP RATIO – A MULTI-COHORT STUDY

PONE-D-22-29652R1

Dear Dr. Lind,

We’re pleased to inform you that your manuscript has been judged scientifically suitable for publication and will be formally accepted for publication once it meets all outstanding technical requirements.

Kind regards,

Xu Chen, Ph.D.

Academic Editor

PLOS ONE
---

## [Editor Report · Acceptance letter]

17 Feb 2023

PONE-D-22-29652R1 

THE METABOLIC PROFILE OF WAIST TO HIP RATIO – A MULTI-COHORT STUDY 

Dear Dr. Lind:

I'm pleased to inform you that your manuscript has been deemed suitable for publication in PLOS ONE. Congratulations! Your manuscript is now with our production department. 

Kind regards, 

on behalf of

Dr. Xu Chen 

Academic Editor

PLOS ONE